# Metabolome Features of COPD: A Scoping Review

**DOI:** 10.3390/metabo12070621

**Published:** 2022-07-05

**Authors:** Suneeta Godbole, Russell P. Bowler

**Affiliations:** 1Department of Biostatistics and Informatics, Colorado School of Public Health, University of Colorado, Anschutz Medical Campus, Aurora, CO 80045, USA; 2Division of Medicine, National Jewish Health, Denver, CO 80206, USA; bowlerr@njhealth.org

**Keywords:** chronic obstructive pulmonary disease (COPD), metabolomics, emphysema, exacerbations

## Abstract

Chronic obstructive pulmonary disease (COPD) is a complex heterogeneous disease state with multiple phenotypic presentations that include chronic bronchitis and emphysema. Although COPD is a lung disease, it has systemic manifestations that are associated with a dysregulated metabolome in extrapulmonary compartments (e.g., blood and urine). In this scoping review of the COPD metabolomics literature, we identified 37 publications with a primary metabolomics investigation of COPD phenotypes in human subjects through Google Scholar, PubMed, and Web of Science databases. These studies consistently identified a dysregulation of the TCA cycle, carnitines, sphingolipids, and branched-chain amino acids. Many of the COPD metabolome pathways are confounded by age and sex. The effects of COPD in young versus old and male versus female need further focused investigations. There are also few studies of the metabolome’s association with COPD progression, and it is unclear whether the markers of disease and disease severity are also important predictors of disease progression.

## 1. Introduction

Chronic obstructive pulmonary disease (COPD) is a heterogeneous disease, currently defined by airflow obstruction but consisting of a constellation of phenotypes, primarily chronic bronchitis and emphysema. Although consistent with the definition of COPD, chronic bronchitis, chronic airflow obstruction, and emphysema have distinct research definitions and are assessed using medical history (chronic productive cough), lung function (post-bronchodilator spirometry), and imaging (CT scans), respectively. Despite this, the spirometric definition of COPD (post-bronchodilator FEV_1_/FVC < 0.7) is required for a formal diagnosis of COPD even though it may result in under-diagnoses of COPD in the young and over-diagnoses of COPD in the elderly [1].

Clinically important features that are relevant to understanding the metabolomic associations of COPD include a predominant diagnosis in the elderly, sexual dimorphisms, and associations with non-pulmonary diseases such as cardiovascular disease, osteoporosis, depression, sarcopenia, and extrapulmonary cancers. Tobacco smoking is the most important risk factor in developed nations, and it is important to consider current smoking because of its profound effect on the metabolome [2]. Some metabolomics literature includes comparisons of COPD to never-smokers (<100 cigarettes in a lifetime) which should be distinguished from comparisons to former smokers. Another important design feature is whether studies are case-control or population-based.

Because COPD has systemic manifestations, plasma and serum are most frequently profiled in metabolomics studies. Lung-specific biosampling of tissue by surgery or airway epithelial lining fluids (ELF) by bronchoscopy (bronchoalveolar lavage fluid) are the only direct methods to sample the lung but are associated with a high risk of complications, which often precludes their use in large epidemiological studies. Other indirect methods to sample the lung include expectorated or induced sputum or exhaled breath (direct volatile organic compounds (VOCs) or condensates). One publication reported on the stool metabolome in COPD [3] and three reported on urine [4,5,6].

Nearly all traditional metabolome profiling methods have been tried in COPD, with targeted or untargeted mass spectrometry techniques being the most common. A common problem in the metabolomic literature on COPD is how to standardize measurements. For instance, sampling lung ELF by bronchoscopy with normal saline lavage (BAL) typically dilutes the ELF 100-fold. There are also no clear methods to standardize measurements from sputum (also frequently induced by hypertonic saline), exhaled breath, or lung tissue, which is significantly less cellular or dense in patients with emphysema. Additional technical challenges have been described in [7].

In this review, we comprehensively evaluate the COPD metabolome primary literature with regard to airflow limitation, chronic bronchitis, and emphysema. We evaluate the evidence for the classes of metabolites that are most often associated with COPD as well as the impact of important factors that may explain these associations (smoking, gender). These studies have repeatedly identified certain classes of metabolites such as sphingolipids (ceramides, sphingomyelins), amino acids, and energy-related pathways (carnitine) that are associated with COPD.

## 2. Results

The 37 articles included in this review spanned a wide range of phenotypic presentations of COPD from airflow obstruction operationalized by FEV_1_, FEV_1_/FVC, and FEV_1_ percent predicted to emphysema, exacerbation frequency and severity, cachexia, chronic bronchitis, DLCO, TLC, and CoHB. Most publications identified COPD cases vs. healthy current-, former-, or never-smoker controls. Several studies focused on COPD-enriched cohorts such as COPDGene [8], SPIROMICS [9], ECLIPSE [10], or the Karolinska COSMIC cohort [11]. Two of the studies used population cohorts from Europe and the United States, and most of the studies (22) used case-control designs with either age- and sex-matched controls or healthy controls. The metabolomics characterization was conducted on a diverse set of biological samples including serum, plasma, BALF, lung tissue, sputum, exhaled breath condensate (EBC), urine, and feces. One key distinction between serum and plasma was the use of fasting or non-fasting samples as this would affect the metabolomics characterization. Six of the serum studies used fasting draws, whereas the other eight did not specify the blood draw regime; similarly, three of the plasma studies specified the use of fasting blood draw, whereas five were from the COPDGene cohort that did not require participants to be fasting at the blood draw. Figure 1 depicts altered pathways in COPD by biospecimen types and sex-stratified analyses.

### 2.1. Metabolome Features of COPD vs. Controls

#### 2.1.1. COPD Status Based on Spirometric Assessment of Lung Function

Out of the 37 studies that met the inclusion criteria for this review, 28 contained analyses related to the identification of COPD versus smoker and non-smoker controls (post-bronchodilator FEV_1_/FVC ≥ 0.7). The two largest studies in this category, Prokic et al. (2020) and Yu et al. (2019), used population cohorts in their analyses.

Prokic et al. [12], used nuclear magnetic resonance (NMR) to measure 161 plasma metabolites in 4948 subjects from the Rotterdam Study (RS) and 609 subjects from the Erasmus Rucphen Family study (ERF) with a replication meta-analysis in 717 subjects from the Lifelines-DEEP study (LLDEEP), two cohorts of the FINRISK study (*n* = 11,498), and the Prospective Investigation of the Vasculature in Uppsala Seniors (PIVUS) study (*n* = 854). In this study, the glycoprotein acetyl A1 (GlycA) level was higher in COPD patients and was a predictive biomarker of COPD incidence. Other metabolites were not significantly associated with phenotypes after adjustment for covariates and multiple testing [12].

Yu et al. [13], used the Metabolon platform to generate serum metabolomes from 2354 African Americans and 1529 European-Americans from the Atherosclerosis Risk in Communities (ARIC) study and 859 Europeans from the Cooperative Health Research in the Augsburg Region (KORA) study. They found 17 metabolites associated with COPD cases vs. control status at an FDR < 0.05 [13]. Significant metabolites included amino acids and lipids. COPD was associated with increases in 3-(4-hydroxyphenyl)lactate, 3-methoxytyrosine, homocitrulline, ornithine, succinylcarnitine, oleoylcarnitine, 5-dodecenoate (12:1n7), 7-alpha-hydroxy-3-oxo-4-cholestenoate (7-Hoca), glycerol, pseudouridine, theophylline, and 1-methylurate and decreases in serotonin (5HT), glycerate, docosahexaenoate (DHA, 22:6n3), and androsterone sulfate [13].

#### 2.1.2. Evidence from Other Smaller Studies of Metabolite Classes Associated with COPD

##### Carnitines

Similar to the large population cohort studies described above, Kim et al. [4], used liquid chromatography and mass spectrometry (LC-MS) on serum and urine samples in a study of 59 TB-related COPD patients, 70 smoking-related COPD patients, and 39 healthy controls (including never-smokers and smokers) and found differences in the acylcarnitines. Similar results were shown in a COPDGene discovery cohort of 839 and a SPIROMICS replication cohort of 446 using LC-MS on serum samples [14]. The differences in carnitines in COPDGene and SPIROMICS were more pronounced in women compared to men. In another study by Naz et al. of untargeted serum metabolomics in 38 never-smokers, 40 smokers without COPD, and 37 smokers with COPD, medium- to long-chain carnitines were significantly downregulated in female COPD patients versus female smokers without COPD [15]. Carnitines have also been identified by partial least squares-discriminant analysis (PLS-DA) in a serum study of 30 COPD patients and 30 healthy former and never-smoker controls by direct infusion mass spectrometry using a hybrid triple-quadrupole-time-of-flight mass spectrometer (DI-ESI-QTOF-MS) on the serum samples [16].

**Figure 1 metabolites-12-00621-f001:**
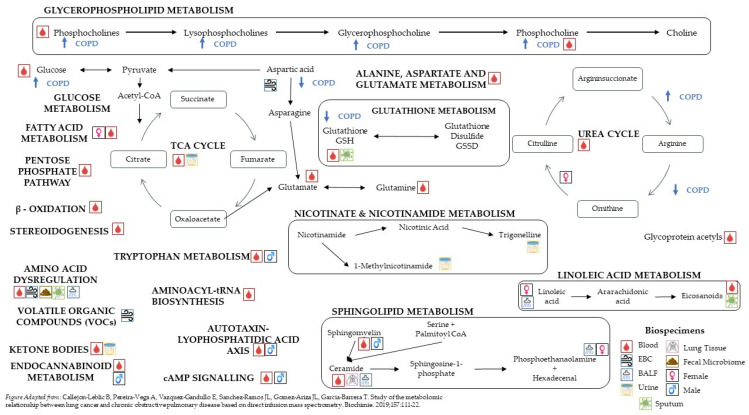
Pathways affected by COPD status [16].

##### Lipids (Sphingolipids: Ceramides, Glycophospholipids)

Many studies have reported associations between lipid classes and COPD in multiple compartments such as lung tissue, bronchoalveolar lavage fluid (BALF), and blood. For instance, in a study of 69 subjects with COPD and 16 smoker controls using LC-MS, the total ceramides were lower in the lung tissue of severe COPD patients, but slightly higher in mild-to-moderate COPD; however, sphingosine-1-phosphate (S1P) was higher in the lungs of severe COPD patients [17]. The correlation of ceramide with caspase 3 suggested a possible role for ceramides in mediating early lung destruction [17]. Other smaller serum studies showed an association between smoking and a decrease in ceramides in serum, pathway enrichment of fatty acids and sphingolipids in females, and a decrease in sphingomyelins in COPD subjects [15,18,19]. Additionally, Gillenwater et al. found ceramide (d18:1/17:0, d17:1/18:0)* and N-stearoyl-sphingosine (d18:1/18:0)* to be associated with COPD status particularly in males [14]. Using a weighted correlation network analysis (WGCNA), they found a network module of ceramides and sphingomyelins that was strongly associated with COPD status in unstratified and also in a sex-stratified analysis, particularly in males. They also found diacylglycerols and phosphatidylethanolamines (PE) to have a lower abundance in COPD cases. The COPD associations with glycophospholipids and long-chain fatty acids were also supported by Kim et al. [4], as well as by Titz et al. in a serum lipidome study using 4 MS platforms of 39 mild/moderate COPD patients, 39 healthy smokers, 38 former smokers without COPD, and 39 never-smokers, in which smoking and COPD, especially, were associated with an increase in glycero(phospho)lipids, including triglycerols, a decrease in polyunsaturated fatty acids, an increase in monounsaturated fatty acids, and an imbalance in eicosanoids [18]. In a smaller study of 25 COPD patients and 21 never-smoker controls, Kilk et al. found that phosphatidylcholines (PCs) were generally decreased in patients with COPD; however, only the differences for the polyunsaturated fatty-acid (PUFA)-containing PCs were statistically significant [19]. Lysophosphatidylcholines showed a tendency towards accumulation in COPD patients [19]. Findings are also supported by a small study of 20 COPD patients and 5 smoking controls with normal lung function, which showed a COPD association with sphingolipids (e.g., sphingomyelins) and phospholipids [20]. These investigators found that ratios such as lysophosphatytilinositol (I) to cholesterol ester (CE) were lower in COPD groups [20].

##### Amino Acids

There have also been many reports of associations between amino acids and COPD. In an NMR study of serum in 79 COPD patients, 59 control smokers, and 7 never-smokers, creatine, glycine, histidine, and threonine concentrations were reduced in COPD patients compared to non-COPD smokers [21]. Histamine was higher in COPD patients compared to non-COPD smokers [21]. Kim et al. found differences in long-chain fatty acids and amino acids between smoking-related COPD and healthy controls, which included never-smokers and smokers [4]. In a study of serum from 30 COPD patients and 30 former and never-smoker healthy controls, there were 41 metabolites that helped identify COPD by PLS-DA [16]. These included decreased glutamate but elevated arginine and phenylalanine, among others [16]. There were also elevated lysophosphocholine, phosphocholine, and triacylglycerides (TAG) [16].

Branched-chain amino acids (BCAAs) have also frequently been reported to be associated with COPD. In an NMR study of serum and urine samples from 32 COPD patients and 21 smokers and never-smoker controls, COPD patients had decreased lipoprotein and amino acids, especially BCAAs and increased glycerophosphocholine in serum [6]. In the same study, it was noted that the urine of COPD patients had decreased 1-methylnicotinamide and creatinine, and increased acetate, ketone bodies, carnosine, m-hydroxyphenylacetate, phenylacetyglycine, pyruvate, and alpha-ketoglutarate [6].

Zheng et al., in an NMR study using the fasting serum of 54 COPD patients and 74 healthy current-, former-, and never-smoker controls from Wenzhou, China, reported that leucine and valine have lower concentrations in COPD patients [22]. In a study of 66 current- and former-smoker controls and 178 COPD patients from the ECLIPSE cohort, serum measured by NMR showed decreased N,N-dimethylglycine and increased glutamine, phenylalanine, 3-methylhistidine, and ketone bodies in COPD patients [23]. Compared to less severe COPD, GOLD stage IV patients showed decreased BCAAs [23]. Bertini et al. showed a decrease in BCAAs in COPD patients compared to healthy non-obstructed, non-current-smoker controls in a study of exhaled breath condensate in 62 subjects (37 COPD, 25 controls) [24]. Lower plasma amino acids have also been reported in COPDGene and SPIROMICS [14].

In a study of 133 patients with COPD and 101 normal pulmonary function controls from Peking University, China, changes in the amino acid metabolism pathways that suggest a protein breakdown were reported in COPD patients [25]. Gillenwater et al. reported COPD associations with glutamate; arginine and proline metabolism; lysine metabolism; methionine; cysteine; SAM and taurine metabolism; and phenylalanine metabolism [26]. In the ECLIPSE cohort, severe COPD, cachexia, and emphysema had higher glutamine, aspartate, and arginine levels and lower aminoadipate [27].

In a study of plasma from 60 severe COPD patients and 30 current- and former-smoker controls with no lung disease from Boston, LC- and GC-MS revealed reductions in BCAAs and an elevation in lactate, fructose, and five-carbon sugars in the non-survivors (died within 3 years) [28]. These findings were generally confirmed in 30 independent subjects from another cohort in Tenerife [28].

##### TCA Cycle

Related to changes in amino acid metabolism are changes in energy production or the tricarboxycyclic acid cycle (TCA) cycle. In a study of 20 healthy controls (including some smokers) and 140 COPD patients who had untargeted metabolomics on fasting serum, the anaerobic glycolysis pathways were increased and there was dysregulation of the TCA cycle (e.g., increased maleate, pyruvate, and lactate in COPD patients) [29]. More severe COPD (e.g., GOLD Stage 4) had the most severe disturbances [29]. There were trends toward more dysregulation in elderly COPD versus younger COPD patients [29]. TCA disruptions were also seen in an untargeted serum metabolomics study of 38 never-smokers, 40 smokers without COPD, and 37 smokers with COPD [15]. In the Boston and Tenerife studies of COPD survival, non-survivors had significant increases in TCA cycle intermediates such as alpha-ketoglutarate, succinate/succinylcarnitine, succinate, fumarate, and malate, suggesting stressed energy needs [28]. These findings were supported by network modules [14] and Naz et al., where enriched pathways for both sexes included the TCA cycle [15].

##### Lipoproteins

In a COPD study from Wenzhou, China, Zheng et al. found lipoproteins, mainly low-density lipoproteins (LDL), very low-density lipoproteins (VLDL), and N-acetyl-glycoprotein to have the largest differences between COPD patients and current-, former-, and never-smoker controls, with the former having lower values [22]. Ubhi et al. also found negative associations between N,N-dimethylglycine, LDL/VLDL, polyunsaturated lipid, O-acetylated glycoproteins, and COPD status in the ECLIPSE cohort, using NMR on serum in 15 non-smokers, 53 smokers, and 163 COPD patients [23]. Of these metabolites, only N,N-dimethylglycine had stronger associations in GOLD IV patients compared to smoking and non-smoking controls [23].

##### Other Small Molecules

In a study of eicosanoids from the induced sputum of 76 COPD and 37 current-, former-, and passive-smoker controls, concentrations of LTE4, LTD4, PGD2, PGE2, 8-iso-PGE2, 8-iso-PGF2a, 5-oxo-ETE, 12-oxo-ETE, and 11-dehydro-TBX2 were significantly higher in COPD subjects compared to healthy controls, whereas concentrations of tetranor-PGE-M and tetranor-PGD-M were lower [30]. There were no differences in eicosanoids or eosinophilia between COPD patients treated with inhaled corticosteroids (ICS) compared to those with bronchodilators only [30]. In a serum study of healthy smokers (*n* = 37), COPD smokers (*n* = 41), and non-smokers (*n* = 37), Chen et al. identified 17 metabolites that were differentially expressed in COPD smokers compared to both healthy smokers and healthy non-smokers, but that were not significantly different between healthy smokers and non-smokers [31]. Five of the seventeen were unknown peptides and further analysis showed that one of these peptides was the N-terminal-modified and C-terminal-truncated fibrinogen peptide B (mFBP). Previous literature has shown an association of plasma fibrinogen with a reduced pulmonary function and an increased risk of COPD [31].

#### 2.1.3. Metabolomic Signatures of COPD in Other Compartments

Although the majority of COPD metabolomics studies have been in serum or plasma, biofluids from other sites have been studied such as BALF, exhaled breath (EB), exhaled breath condensates (EBC), sputum, and even stool.

##### BALF

In a study of BALF and serum from 40 healthy never-smokers, 39 smokers without COPD, 35 subjects with COPD from the COSMIC cohort, and a validation cohort including 16 current smokers without COPD and 11 with COPD, the main differentiators were in female smokers with normal lung function and COPD patients. They included nine BALF lipid mediators: 9,10,13-TriHOME (9,10,13-trihydroxy-11E-octadecenoic acid), 12(13)-EpOME (12[13]epoxy-9Z-octadecenoic acid), 9(10)-EpOME (9[10]-epoxy-12Z-octadecenoic acid), 9,10-DiHOME (9[10]-dihydroxy-12Z-octadecenoic acid), 12,13-DiHOME (12[13]-dihydroxy-12Z- octadecenoic acid), 12-HHTrE (12-hydroxy-5Z,8E,10E-heptadecatrienoic acid), 5-KETE (5-oxo-ETE, 5-oxo- 6E,8Z,11Z,14Z-eicosatetraenoic acid), TXB2 (thromboxane B2), and 9-KODE (9-oxo-10E,12Z-octadecadienoic acid) [32]. In serum, the differences included the abundances of 5-LOX products and increases in CYP-derived (5[6]-EpETrE, 11[12]-EpETrE), and putative platelet-derived products (12-HETE, 12-HHTrE) in COPD patients, but there were no gender differences [32].

##### EB, EBC, and eNOSE

In an exhaled breath (EB) study of 27 COPD patients and 7 healthy subjects measuring 37 VOCs, 2 were higher in COPD (decane and decane, 6-ethyl-2-methyl) and 7 were lower in COPD (benzene, butylated hyroxytoluene, hexan, hexyl ethlyphosphonofluoridate, limonine, pentene, and propanol) [33]. Similar hydrocarbon signatures were seen in a study of 190 subjects including 89 with COPD, who had 134 VOCs measured in EB [34]. In the study, seven VOCs were associated with COPD across two sites; vinyl acetate, benzene, toluene, m,o-xylene, 1-ethyl-3-methyl benzene, and 1,6-dimethyl-1,3,5-hepatriene were positively associated with COPD status, whereas indole was negatively associated [34]. In a study of exhaled breath condensate (EBC) from 35 patients with COPD and 35 healthy volunteers, cyclohexanone was higher in COPD patients [35]. In another EBC study from 22 COPD patients and 14 controls, 14 features were found to distinguish between the groups with an over-representation of fatty acids, aldehydes, and amino acids [36]. In a study of EB collected by eNOSE in 23 COPD patients and 33 healthy controls, there was an increase in the abundance of alpha-pyrene, acetaldehyde, 2-butyloctanol, octane, methyl isobutyrate, butanal, 2-propanol, 3-hexanone, cyclopentanone, and 3-methyl-propanal, as measured by the area under the chromatograms, in the exhaled breath of patients with COPD compared with healthy volunteers [37]. Additionally, the authors found decreases in the abundance of delta-dodecalactone, 2-methyl butanoic acid, 2- acetylpyridine, tetradecane, [E]-cinnamaldehyde, and vinylpyrazine.

##### Oxidative and Nitrative Stress Markers

Sputum supernatant metabolites from 980 subjects in the SPIROMICS cohort (77 healthy non-smokers [NS], 341 smokers with preserved spirometry [SPS], and 562 COPD subjects) showed that the most significant differences between subjects with multiple exacerbations compared to those with none (at an FDR < 0.1) were sialic acid, hypoxanthine, xanthine, methylthioadenosine, adenine, and glutathione [26]. Evidence for the dysregulation of nitric oxide and the arginine pathways (nitrative stress) was also found more in plasma from women than from men in the COPDGene and SPIROMICS studies [15].

##### Fecal Metabolome

In a study of the fecal microbiome from 28 COPD patients and 29 controls, there were no significant differences in the fecal metabolome; however, sixteen metabolites, all from the lipid, amino acid, or xenobiotic classes, were identified as significantly different between COPD and healthy samples following adjustments for covariates (age, sex, and BMI) using a linear model, with generally lower levels in the COPD patients [3]. Some of these metabolites were associated with specific microbial species in the microbiome analysis [3]. Additionally, N-acetyl-cadaverine, N-acetylglutamate, suberate (C8-DC), undecanedioate (C11-DC), harmane, and N-carbamoylglutamate were identified in COPD-associated network models of fecal microbiome and metabolites [3]. Using NMR, McClay et al. found differences between 197 COPD subjects and 195 current- and never-smokers in hippurate and formate in urine samples, and they suggested that both of these metabolites are associated with gut microflora and may indicate differences in microflora by lung function [5].

### 2.2. Reduced Lung Function

Several studies examined the associations between metabolites and lung function, which was operationalized as FEV_1_, FVC, FEV_1_ percent predicted, and FEV_1_/FVC ratio. In a large cross-sectional study (*n* = 4742) of three population cohorts with 393 COPD cases, Yu et al. found 95 serum metabolites associated with FEV_1_ and 100 serum metabolites associated with FVC at an FDR < 0.05, with the top five metabolites associated with FEV_1_ being the positively associated glycerine, 3-phenylpropionate, 2-methylbutyrylcarnitine (C5), and the negatively associated 3-(4-hydroxyphenyl)lactate, and asparagine [13]. Twenty-three metabolites were associated with both FEV_1_ and FVC, including negative associations for the lipid, glycerol, and the nucleotide, N2,N2-dimethylguanosine, and positive associations for three carbohydrates and several gamma-glutamyl amino acids [13]. There were also eight amino acids and their derivatives associated with FEV_1_ and FVC but these were both positively and negatively associated [13]. Among the 30 novel metabolites, they found associated with FEV_1_, some have been previously linked to metabolic health including BCAAs and their byproducts, aromatic amino acids, glycine, and glutamate/glutamine. Yu et al. also found 10 metabolites associated with FEV_1_/FVC at an FDR < 0.05 in serum [13]. In their analysis, several pathways were enriched including aminoacyl-tRNA biosynthesis, phenylalanine metabolism, nitrogen metabolism and alanine, aspartate, and glutamate metabolism for FEV_1_ [13]. Similar to this population study, Diao et al. found creatine, histidine, threonine, lactate, proline, and serine metabolites to be positively correlated with FEV_1_ percent predicted in a cross-sectional case-control study [21].

In contrast, studies enriched with COPD patients have found a dysregulation in other pathways. In a sample of 957 non-Hispanic Whites (NHW) from the COPDGene cohort, Gillenwater et al. found that 99 metabolites and 30 metabolite subclasses were significantly associated with FEV_1_/FVC and that 79 metabolites and 23 metabolite subclasses were significantly associated with FEV_1_ percent predicted [38]. Among the pathways, diacylglycerols, gamma-glutamyl amino acids, sphingomyelins, and lipids were found to be associated with airflow obstruction [38]. The pathways associated with FEV_1_/FVC included glycophosphatidylinositol, propionylcarnitine (C3), and ergothioneine, a xenobiotic. Enrichment analysis found metabolites in the diacylglycerol and BCAA (leucine, isoleucine, and valine) sub-pathways to be enriched for associations with FEV_1_/FVC [38]. In a study of 131 subjects from COPDGene, there were 269 metabolites, particularly glycerophospholipids, associated with FEV_1_/FVC [39]. In a smaller study of 41 COPD smokers, 37 non-smokers, and 37 healthy smokers, using Pearson correlations on LC-MS data from fasting serum, Chen et al. found a negative correlation between four peptide biomarkers and FEV_1_/FVC and DLCO, but no association with total lung function (TLC) and carboxyhemoglobin (CoHB) was also observed [31]. There was also a significant correlation between non-peptide small molecule metabolites and each of the four lung function parameters described above [31]. In an NMR study of 197 COPD patients and 195 current- and never-smoker subjects without COPD, trigonelline, a breakdown product of niacin, was positively associated with FEV_1_ even after adjustments for creatinine in the urine samples [5]. One study examining the sex-specific differences in COPD patients in untargeted serum metabolomics of 38 never-smokers, 40 smokers without COPD, and 37 smokers with COPD showed that lysophosphatidic acid (lysoPA) was most strongly positively correlated with FEV_1_ percent predicted but was only correlated with FEV_1_ in males [15].

Finally, in a study of untargeted metabolomics in plasma and BALF from 131 subjects in the SPIROMICS cohort, there were more and stronger associations with FEV_1_/FVC in BALF and emphysema compared to plasma [40]. The BALF compounds most strongly associated with FEV_1_/FVC included one nicotine metabolite, p-cresol, four phosphatidylethanolamines, four phosphatidylcholines, two cardiolipins, free homocysteine, one bile acid, one sphingolipid, one cysteine-derived compound, one glycine-derived compound, one threonine-derived compound, one sphingomyelin, two glycerolipids, and two likely xenobiotics [40]. The BAL compounds associated with FEV_1_/FVC were enriched for multiple compound classes such as amino acid-containing compounds, fatty acids, and phospholipids including lysophospholipids, phosphatidylethanolamines, phosphatidylinositols, phosphatidylcholines, and phosphatidylserines [40]. Current smoking and neutrophilia had a large impact on the metabolites in the BALF, whereas age and sex were more strongly associated with the metabolites in the plasma [40].

### 2.3. Emphysema

Publications that examined the association between metabolites and emphysema and met the criteria for this review were mainly drawn from the COPDGene study. Using 1008 participants from the COPDGene study with plasma-based proteomic and metabolomic data, Mastej et.al. explored networks related to emphysema and lung function. The protein–metabolite network associated with emphysema as a percentage of total lung volume on chest-computed tomography consisted of 13 proteins and 10 metabolites, which included 1-stearoyl-2-linoleoyl-GPI (18:0/18:2), androsterone glucuronide, 1-stearoyl-2-docosahexaenoyl-GPE (18:0/22:6), 1-palmitoyl-2-docosahexaenoyl-GPE (16:0/22:6), 1-palmitoyl-2-linoleoyl-GPI (16:0/18:2), 1-ribosyl-imidazoleacetate, valine, palmitoyl-linoleoyl-glycerol (16:0/18:2) [2], 1-stearoyl-2-arachidonoyl-GPI (18:0/20:4), and glutamate [41]. In the same samples from COPDGene, but only using 957 non-Hispanic Whites (NHW), Gillenwater et al. found the tricarboxylic cycle metabolite, citrate, and six metabolite-network modules significantly associated with the percentage of emphysema [38]. In a smaller study of 129 subjects with MS-targeted sphingolipids and 131 subjects from COPDGene with untargeted MS plasma profiling, sphingomyelins and ceramides were inversely associated with emphysema. Oxidative phosphorylation pathways were over-represented in the emphysema phenotype, which may be explained by the oxidizing nature of cigarette smoke [39]. In another study from the COPDGene sample (*n* = 839), but adding a demographically and functionally similar pool from SPIROMICS (*n* = 446), Gillenwater et al. explored sex-specific associations between metabolites and the percentage of emphysema and found network modules for (1) amino acids, bile acids, acyl cholines, lysophospholipids; (2) xenobiotics, amino acids, and the TCA cycle; and (3) steroids [14]. Of these modules, the first two were significant in males, whereas the third was significant in females; however, these associations found in the COPDGene cohort were not statistically significant in the SPIROMICS cohort [14]. Similarly, in serum, higher glutamine, aspartate, arginine, and lower aminoadipate were associated with emphysema in the ECLIPSE cohort [27]. In an NMR study of serum samples from 157 subjects in the SPIROMICS cohort, there were no metabolites associated with FEV_1_ or emphysema; however, the cohort in this study has a lower severity of airflow obstruction compared to COPDGene [42]. In contrast, in the SPIROMICS cohort, but using BAL samples from 115 subjects, Halper-Stromberg et al. found 791 compounds associated with the percentage of emphysema and an enrichment of several classes including amino acid-derived compounds, fatty acids, carnitines, and phospholipids including phosphatidylethanolamines, phosphatidylinositols, and phosphatidylcholines [40]. Diao et al. found that creatine, histidine, 3-hydroxybutyrate, betaine, carnitine, glutamine, acetyl carnitine, and valine metabolites were correlated with emphysema in a study of 138 participants using NMR analysis of fasting serum samples [21].

Two studies examined the association between metabolites and COPD status using an emphysema-based definition of COPD. Emphysema was quantified on CT scans as the percentage of low attenuation pixels (LAA%), with attenuation values of <−950 Hu. Subjects with >20 LAA% were classified as having COPD [27]. Both studies were conducted by Ubhi et al. in the ECLIPSE cohort. Ubhi et al. used fasting serum samples from 77 emphysematous and 41 non-emphysematous patients from the ECLIPSE cohort to show the associations between metabolites and emphysema using NMR. They found significant increases in 3-methylhistidine, glutamine, phenylalanine, and 3-hydroxybutyrate along with a decrease in the metabolites associated with creatine biosynthesis/glycine degradation, BCAA degradation, lipid metabolism, and post-translational glycoprotein acetylation [23]. In another analysis of the ECLIPSE cohort, when comparing emphysematous to non-emphysematous patients, Ubhi et al. found increases in glutamine, serine, histidine, arginine, proline, asparagine, aspartic acid, glycine–proline, and lysine, whereas the concentrations of tryptophan, sarcosine, β-aminoisobutyric acid, and aminoadipic acid were decreased [27].

### 2.4. Exacerbation Frequency and Severity

Using a sample of 129 current and former smokers from COPDGene, Bowler et al. found that ceramides (e.g., trihexosylceramides) were positively associated with exacerbations and that sphingosine-1-phosphate was negatively associated with exacerbations in targeted mass spectrometry of plasma [43]. Many of these associations were similar to the subjects with untargeted profiling in the same cohort [43]. Plasma amino acids, carnitines, and carbohydrates were associated with a history of frequent or severe COPD exacerbations in another analysis of 149 participants of the COPDGene cohort [39]. A history of severe exacerbations is associated with dysregulated energy pathways and more degradation pathways, particularly carbohydrate degradation, nucleoside degradation, fatty-acid degradation, and amino-acid degradation [39]. The degradation of these pathways highlights the systemic effects of COPD since carbohydrate breakdown provides a source of cellular energy for other processes such as protein transport; nucleosides act as signaling molecules or provide chemical energy to cells and fatty acids store energy in the cell [39]. Esther et. al. found that sialic acid and hypoxanthine were strongly associated with measures of disease severity and that the elevation of these biomarkers was associated with a shorter time to exacerbation and improved prediction models of future exacerbations in sputum samples from the SPIROMICS cohort [6]. In a sample of 957 NHW from the COPDGene cohort, N,N,N-trimethyl-alanylproline betaine (TMAP) was significantly associated with exacerbation frequency [38]. In a serum NMR study of 157 subjects from the SPIROMICS cohort, 20 out of 27 metabolites (mostly amino acids, including tryptophan, and BCAAs) were lower in those with at least one exacerbation during the preceding year [42]. In the Peking University China study, there was a subset of the matched study of 48 healthy controls, 48 stable COPD patients, and 48 patients with an acute exacerbation of COPD (AECOPD), in which glutamylphenylalanine and taurine were lower with AECOPD [25].

### 2.5. Other Outcomes

Other phenotypes examined in these publications included chronic bronchitis, and cachexic; however, there is minimal evidence for each outcome. In a sample of 957 NHW using plasma, Gillenwater et al. found no metabolites significantly associated with chronic bronchitis [38]. However, Esther et. al. found that sialic acid was elevated in the sputum samples of chronic bronchitis patients compared to patients without chronic bronchitis [26].

In comparing cachexic and non-cachexic patients, Ubhi et al. showed marked increases in glutamine, phenylalanine, and serine concentrations, whereas other metabolite increases included histidine, arginine, proline, asparagine, aspartic acid, citrulline, and valine. Additionally, they found decreases in aminoadipic acid and hydroxylysine [27]. In a separate analysis, Ubhi et al. also separated metabolite dysregulation due to BMI and fat-free mass changes versus COPD in cachexia patients, and they found positive associations with HDL, ascorbate, and N-methylnicotinate and a negative association with glutamine [23]. Additionally, the authors found that BCAAs, their degradation products, 3-methylhistidine, ketone bodies, and triglycerides were negatively correlated with cachexia [23].

#### Metabolite Endotyping

There have been few studies that have used metabolite endotyping in COPD patients. Using a support vector machine with recursive feature selection and clustering, Gillenwater et al. created a 12-metabolite panel that clustered subjects into two COPD clusters with enriched pathways for sphingomyelins [44].

## 3. Discussion

Although many of these cited publications used different definitions of COPD, types of biosamples, and metabolomics techniques, there are some common themes. For instance, many studies identify the dysregulation of carnitines, lipids (especially sphingolipids), amino acids (especially BCAAs), and the TCA cycle. These classes belie the underlying impaired physiology of COPD such as inflammation (ceramides), muscle wasting, and energy metabolism (TCA cycle, amino acids, carnitines). The exhaled breath and exhaled breath condensate studies evaluated a different variety of metabolites than the other studies and consistently identified volatile hydrocarbons, for which these techniques may be more sensitive. There are also differences between the population-based studies, which have much higher control-to-COPD ratios than case-control studies, which are the majority of the smaller published studies. The largest population study found an enrichment of aminoacyl-tRNA biosynthesis, phenylalanine metabolism, nitrogen metabolism, as well as alanine, aspartate, and glutamate metabolism pathways, whereas the COPD-enriched studies found an association between diacylglycerols, gamma-glutamyl amino acids, and lipids.

There are some themes in COPD metabolomics research that need additional investigation and consideration in future studies. Although sex has a strong metabolomic signature, only a few studies used a stratified analysis of males and females. The stratified studies did find important differences such as with lysoPA or carnitines and FEV_1_. Across studies, there were multiple limitations noted. The first was the higher percentage of males in almost every study, ranging from 39.8% to 100%. The disproportionate representation of the male sex limits the generalizability of the studies as well as makes them less powered to examine the sex-specific dysregulation of metabolic pathways.

Additionally, there are multiple collections, types of processing, and analytical decisions that may affect the associations found in the specific studies. For instance, metabolites can be characterized by NMR or MS technologies. With NMR characterization, usually used for a smaller targeted set of metabolites, whereas MS characterization can be used to identify over 1000 metabolites, which in itself may lead to different associations with a phenotype. In this review, 7 out of the 37 manuscripts used NMR, 1 used both NMR and MS and the remaining used MS. Also, as with other omics types, an adjustment of *p*-values to account for multiple comparisons was used in all studies, mainly with a false discovery rate correction. However, the analytics methods ranged from network models to partial least squares (PLS), based on the objectives of each paper, so the underlying assumptions of each analytic method could lead to differing results.

Another confounding variable for COPD metabolomics is age, which has a strong influence on the metabolome and is also strongly associated with COPD. In the manuscripts we reviewed, most studies focused on adults in their middle-to-late 50s and older for the COPD cases; however, sometimes the healthy comparison cohort was much younger, as in Cazzola et al. with a mean age of 72 years for the COPD cases and 27 years for the healthy controls [33], or in Bertini et al. with a mean age of 70 years for the COPD cases and 56 years for the controls [24]. These differences between the COPD cases and the controls makes the interpretation of the noted dysregulated metabolites and pathways difficult because it is hard to separate the differences associated with aging and those associated specifically with COPD. Although age is often included as a linear covariate in modeling COPD, there should be more investigation into which COPD pathways are related to aging and which are not. For instance, the sphingolipid metabolism and carnitine pathways are both associated with aging and COPD [45]. It is unknown whether the pathways of disease progression are different in younger versus older subjects.

Although most COPD studies investigated the spirometry-defined phenotype, there were few studies that investigated emphysema and many of these studies were heavily weighted from having just one cohort (COPDGene). However, the consistent findings in these studies were differences in sex hormones, TCA metabolites, and sphingolipids as well as an over-representation in the oxidative phosphorylation pathways. In both studies by Ubhi et al., the classification of COPD status based on emphysema was examined and a dysregulation of amino acids and branched-chain amino acids was found in both. More importantly, there have been few metabolomics studies of COPD progression. Identifying the metabolomic pathways that predict progression are important, not just for prognosis, but also to identify potential targets for interventional studies that could be capable of disease modification, which has remained elusive in COPD.

## 4. Materials and Methods

Google Scholar, PubMed, and Web of Science searches were performed for keywords “metabolomics” and “COPD”, “Chronic Bronchitis”, or “Emphysema”, searched on 14 February 2022 for Google Scholar and PubMed and 10 June 2022 for Web of Science resulting in a list of 2838 matches (Figure 2) using the Publish or Perish results aggregator [46]. For the Web of Science, the results were limited to manuscripts published on the earlier search date. Searches were limited to records from 1980 or after.

Records from the search underwent three levels of review for the following inclusion/exclusion criteria. Included articles (1) were peer-reviewed and published; (2) were English-language publications; (3) were human subject studies; (4) included healthy controls of never-smokers, former smokers, or current smokers; (5) included at least 20 COPD patients; and (6) analyzed at least 20 metabolites. Records were excluded when they did not meet the inclusion criteria and when (1) data were from an intervention study or therapeutic trial, and (2) no metabolite annotations were used. The first level of review was a title review. SVG reviewed the title of each record, eliminating records that did not meet the inclusion criteria or that met the exclusion criteria. Any records for which the inclusion/exclusion criteria could not be ascertained were kept for further screening leading to 276 articles/abstracts. For the second level of review, the abstracts for these 276 records were then manually reviewed by RPB who excluded an additional 223 articles/abstracts based on the inclusion/exclusion criteria. Finally, the full articles were independently reviewed for inclusion, and data were extracted by SVG and RPB concurrently. Discordance in the assessment was resolved through consultation between the reviewers and a joint review. A total of 37 articles were included in this review.

Along with article identification information such as article title, authors, journal, and year of publication, data extraction for each article entailed basic demographic information including age and sex distribution, smoking status, and cohort identification; methods for characterizing the metabolites, data normalization, and transformation procedures; statistical analysis methods; positively and negatively associated metabolites and metabolite networks; and stratified results by sex and any identified pathways.

To synthesize the results from these articles, the results were grouped by outcomes such as COPD case identification, airflow obstruction, or emphysema, and then within each outcome group, the results were grouped by the metabolites identified and associated with the phenotype. Data were sought for airflow obstruction, emphysema, exacerbation frequency and severity, and cachexia, as well as DLCO, TLC, and CoHB. Study summaries and the top 5 positively and negatively associated metabolites and identified pathways are presented in Table 1, while all associated metabolites are presented in Appendix A.

## Figures and Tables

**Figure 2 metabolites-12-00621-f002:**
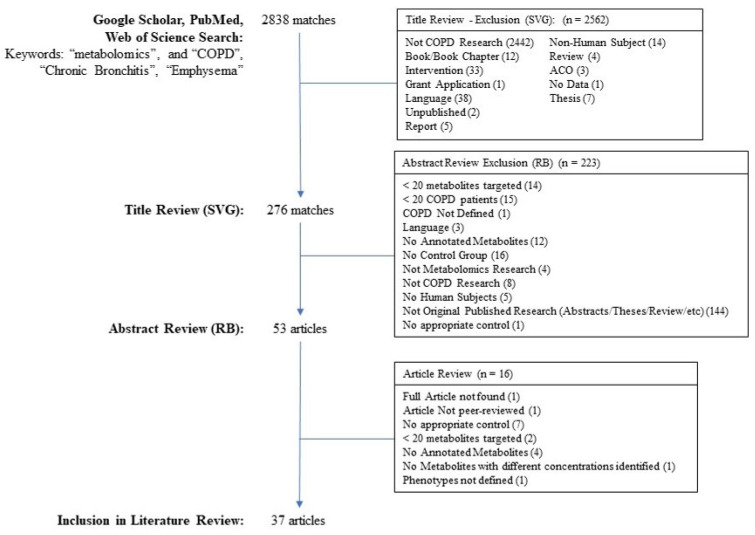
Inclusion/Exclusion Flowchart.

**Table 1 metabolites-12-00621-t001:** Top 5 Positive and Negative Metabolomics Associations by COPD Outcome for Each Study.

Outcome	Authors	Title	Sample Description	Analytic Platform	Sample Type	Top 5 Positive Metabolite Associations	Top 5 Negative Metabolite Associations	Pathways Identified
COPD vs. Healthy Control	Bertini et al.	Phenotyping COPD by 1H NMR metabolomics of exhaled breath condensate	37 COPD/25 controls 67.7% male 70/56 yrs (COPD/HC)	NMR	EBC	lactate; acetate; propionate, serine, proline; tyrosine	acetone; valine; lysine	
COPD vs. Healthy Control	Bowerman et al.	Disease-associated gut microbiome and metabolome changes in patients with chronic obstructive pulmonary disease	28 COPD/29 HC 39.9% male 67/60 yrs (COPD/HC)	LC-MS	Feces	N-acetyl-cadaverine; N-acetyltaurine; cotinine; N-carboxymethylalanine; asmol	N-acetylglutamate; 6-oxopiperidine-2-carboxylate; N-acetyltaurine; N-acetylproline; gamma-glutamylglutamate	
COPD vs. Healthy Control	Bregy et al.	Real-time mass spectrometric identification of metabolites characteristic of chronic obstructive pulmonary disease in exhaled breath	22 COPD/14 controls 61.1% male 58.6/58.1 yrs (COPD/HC)	SESI-HRMS	EBC	2-hydroxyisobutyric acid; Aspartic acid semialdehyde; Acetohydroxybutanoic acid; 2-oxoglutaric acid semialdehyde	Pyridine; 11-hydroxyundecanoic acid; (+)-γ-hydroxy-L-homoarginine; Oxo-tetradecenoic acid; Hexadecatrienoic acid	
COPD vs. Healthy Control	Callejon-Leblic et al.	Study of the metabolomic relationship between lung cancer and chronic obstructive pulmonary disease based on direct infusion mass spectrometry	30 COPD/30 HC/30 Lung Cancer 71.1% male 67/56/66 yrs (LC/HC/COPD)	DI-ESI-QTOF-MS	Serum	Acetic acid; Adenine; Dopamine; Phenylalanine; Arginine	Pyroglutamate; Aspartic acid; Creatine; Ornithine; Glutathione	
COPD vs. Healthy Control	Cazzola et al.	Analysis of exhaled breath fingerprints and volatile organic compounds in COPD	27 COPD/7 HC 79.4% male 72/27 yrs (COPD/HC)	Enose GC-MS	EBC	Decane; 6-ethyl-2-methyl-Decane	1,3,5-tri-tert-butyl-Benzene; Butylated hydroxytoluene; 3-ethyl-4-methyl-Hexane; Hexyl ethylphosphonofluoridate; Limonene	
COPD vs. Healthy Control	Celejewska-Wójcik et al.	Eicosanoids and Eosinophilic Inflammation of Airways in Stable COPD	76 COPD/37 HC 68% male 65 years (mean age)	GC-MS HPLC-MS	Sputum	LTE4; LTD4; PGE2; PGD2; 8-izo-PGE2	Tetranor-PGE-M; Tetranor-PGD-M	
COPD vs. Healthy Control	Chen et. al.	Serum Metabolite Biomarkers Discriminate Healthy Smokers from COPD Smokers	41 COPD Smoker/37 Healthy Smoker/37 Healthy Non-Smoker 78% male 39.5/41.8/53.2 yrs (HC/Smoker/COPD)	LC-MS	Fasting Serum	Cotinine; 3-Hydroxycotinine; Unknown 1; Quinic acid; PI(32:2)		
COPD vs. Healthy Control	Esther et al.	Identification of Sputum Biomarkers Predictive of Pulmonary Exacerbations in Chronic Obstructive Pulmonary Disease	SPIROMICS cohort 77 HC/341 smokers preserved spirometry/562 COPD 53% male 55.4/59.6/65 yrs (HC/Smokers/COPD)	UPLC-MS	Sputum	Sialic Acid; Hypoxanthine; Xanthine; Methylthioadenosine; Adenine		
COPD vs. Healthy Control	Gillenwater et al.	Metabolomic Profiling Reveals Sex Specific Associations with Chronic Obstructive Pulmonary Disease and Emphysema	COPDGene Cohort: *n* = 839; 51.7% male; 67 yrs SPIROMICS Cohort: *n* = 446; 52% male; 64.5 yrs	LC-MS (Metabolon)	Plasma	*Network Module (female)*: Ceramides, Sphingomyelins *Network Module (Male)*: Steroids (Androgenic, Pregnenolone, Corticosteroids, Progestin)) *Network Module (Male & COPD)*: Ceramides, Sphingomyelins *Males*: ceramide (d18:1/17:0, d17:1/18:0); octadecenedioate (C18:1-DC); N-stearoyl-sphingosine (d18:1/18:0) *Males & Females*: 4 acyl carnitines.	*Network Module (COPD)*: Diacylglycerols, Phosphatidylethanolamines (PE), Acyl Carnitines *Network Module (COPD)*: Amino Acids, Bile Acids, Acyl Cholines, Lysophospholipids *Males & Females*: retinol (Vitamin A); phosphocholine; ergothionene; 3-formylindole *Male & COPD*: sphingomyelins *Female & COPD*: phosphatidylethanolamines; acyl carnitines *Opposite Direction By Sex*: ceramide (d18:1/17:0, d17:1/18:0)	
COPD vs. Healthy Control	Kilk et al.	Phenotyping of Chronic Obstructive Pulmonary Disease Based on the Integration of Metabolomes and Clinical Characteristics	25 COPD; 21 HC 73.9% male 67/37 yrs (COPD/HC)	Untargeted: Electrospray ionization MS Targeted: HPLC-MS	EBC Serum	Ala; Arg; Gln; Orn; Phe	acylcarnitine C10; acylcarnitine C16:1; acylcarnitine C18; Ser; lysoPC a C18:0	
COPD vs. Healthy Control	Kim et al.	Metabolic Fingerprinting Uncovers the Distinction Between the Phenotypes of Tuberculosis Associated COPD and Smoking-Induced COPD	59 T-COPD (TB)/70 S-COPD 39 healthy controls 100% male 66/68/55 yrs (TB/COPD/HC)	LC-QTOF-MS LC-MS/MS	Plasma Urine	Acylcarnitines C12; Acylcarnitines C141; Acylcarnitines C161; Acylcarnitines C10; Acylcarnitines C121	lysoPC a C182; H1; PC aa C342; Pro; alpha-AAA	
COPD vs. Healthy Control	Liu et al.	Identification of lipid biomarker from serum in patients with COPD	20 COPD/5 Control 60% males 65.8/69.3 yrs (COPD/HC)	ESI-MS	Serum	C16:E1; TAG(54:6); PC(32:1); SM(22:0); ePS(40:5) *Ratios*: C16:1 CE/C19:0 CE; PC(40:4)/ePC(38:2)	ePE(34:2); ePS(38:3); LPE(20:2); PI(36:6); PI(44:6) *Ratios*: PI(38:4)/C16:1 CE; PI(36:2)/C16:1 CE; ePC(38:2)/C16:1 CE; LPC(18:0)/C20:3 CE; LPC(16:1)/C16:1 CE	
COPD vs. Healthy Control	Prokić et al.	A cross-omics integrative study of metabolic signatures of chronic obstructive pulmonary disease	*Discovery*: 4948 Rotterdam Study (44.2% male; 70.3 yrs); 609 Erasmus Rucphen Family study (44.2% male; 49 yrs) *Validation*: 717 Lifelines-DEEP study (43.7% male; 46 yrs); 11,498 FINRISK (47.7% male 49.7 yrs); 854 Prospective Investigation of the Vasculature in Uppsala Seniors (51.8% male; 70 yrs)	NMR	Fasting Plasma	*Discovery*: Glycoprotein acetyls; 3-hydroxybutyrate; Free cholesterol in med. HDL; Acetoacetate *Validation*: Glycoprotein acetyls	*Discovery*: Histidine; Acetoacetate	
COPD vs. Healthy Control	Rodríguez-Aguilar et al.	Ultrafast gas chromatography coupled to electronic nose to identify volatile biomarkers in exhaled breath from chronic obstructive pulmonary disease patients: A pilot study	23 COPD/33 HC 48.2% male 67.7/55.6 yrs (COPD/HC)	eNose (ultrafast GC)	Fasting Exhaled Breath	Alpha-pinene; Acetaldehyde; 2-Butyl octanol; Octane; Methyl isobutyrate	Delta-dodecalactone; 2-Methylbutanoic acid; Indole; 2-Acetylpyridine; Tetradecane; [E]-Cinnamaldehyde	
COPD vs. Healthy Control	Titz et al.	Alterations in Serum Polyunsaturated Fatty Acids and Eicosanoids in Patients with Mild to Moderate Chronic Obstructive Pulmonary Disease (COPD)	39 Never-Smokers/39 Current Smokers/39 COPD/38 Former Smokers 55.5% males 55.8/55.5/57.9/57.1 yrs (NS/S/COPD/FS)	4 MS platforms: shotgun; triacylglycerol; ceramide and cerebroside; eicosanoid lipidomics;	Serum	CE16:1; Cer(d18:1/22:1); DAG 18:1/18:1; DAG 16:0/18:1; 15-HETrE *Clusters*: (HODE); (CE; PC; LPC/LPE [x/20:5]; DHA/EPA)		triacylglycerols (TAGs), diacylglycerols (DAGs), and phosphatidylethanolamines (Pes)
COPD vs. Healthy Control	Wang et al.	Metabonomic Profiling of Serum and Urine by 1H NMRBased Spectroscopy Discriminates Patients with Chronic Obstructive Pulmonary Disease and Healthy Individuals	32 COPD/21 HC 56.6% male 71/63 yrs (COPD/HC)	NMR	Serum	Glycerolphosphocholine	Alanine; Isoleucine; CH3-(CH2)n-HDL; Leucine	
COPD vs. Healthy Control	Wang et al.	Metabonomic Profiling of Serum and Urine by 1H NMRBased Spectroscopy Discriminates Patients with Chronic Obstructive Pulmonary Disease and Healthy Individuals	32 COPD/21 HC 56.6% male 71/63 yrs (COPD/HC)	NMR	Urine	Acetate; Acetoacetate; Acetone; Carnosine; m-Hydroxyphenylacetate	1-methylnicotinamide; Creatinine; Lactate	
COPD vs. Healthy Control	Westhoff et al.	Differentiation of chronic obstructive pulmonary disease (COPD) including lung cancer from healthy control group by breath analysis using ion mobility spectrometry	97 COPD (35 w/Bronchial Carcinoma; 62 w/o)/35 HC *No other details provided*	IMS-MCC	Exhaled Breathe	cyclohexanone		
COPD vs. Healthy Control	Zheng et al.	Predictive diagnosis of chronic obstructive pulmonary disease using serum metabolic biomarkers and least-squares support vector machine	54 COPD/74 HC 60.9% male 71.3/65.1 yrs (COPD/HC)	NMR	Fasting Serum	formate	N-acetyl-glycoprotein, lipoproteins mainly including LDL and VLDL, pUFA, glucose, alanine	
COPD vs. Healthy Control	Zhou et al.	Plasma Metabolomics and Lipidomics Reveal Perturbed Metabolites in Different Disease Stages of Chronic Obstructive Pulmonary Disease	48 HC/48 Stable COPD/48 AECOPD 79.2% male 63.7/67.3/66.2 yrs (HC/S-COPD/AECOPD)	LC-MS	Fasting Plasma	Xanthine; Dimethylglycine; Phenylalanine; D-Alanyl-D-alanine; Cysteinylglycine	Leucine; Oxazepam; L-Tryptopha; Serotonin; gama-Glutamylleucine	Aminoacyl-tRNA biosynthesis; Nitrogen metabolism; valine, leucine and isoleucine biosynthesis; arginine and proline metabolism; glycerine, serine, and threonine metabolism; phenylalanine metabolism; Pantothenate and CoA biosynthesis; Beta-alanine metabolism
COPD (non-survivors) vs. Healthy Control	Pinto-Plata et al.	Plasma metaoblomics and clinical predictors of survival differences in COPD patients	90 COPD/30 Controls 66% males 68.7/68 yrs (COPD/HC)	LC-MS GC-MS	Plasma	2-ethylhexanoate; bradykinin, des-arg (9); Hexadecanedioate; Fucose; HWESASXX*	Benzoate; 2-aminobutyrate; dehydroisoandrosterone sulfate (DHEA-S); caproate (6:0); Isovalerate	*COPD Survivors* vs. *Non-Survivors*: Glyoxylate and dicarboxylate metabolism; Citrate cycle (TCA cycle)
COPD vs. control	Yu et al.	Metabolomics Identifies Novel Blood Biomarkers of Pulmonary Function and COPD in the General Population	ARIC Cohort (*n* = 2354 African Americans, 1529 European American); 39.8% male; 53.0/54.6 yrs (AA/EA) KORA cohort (*n* = 859 Europeans); 46.8% male; 53.8 yrs	LC-MS	Serum	3-(4-hydroxyphenyl)lactate; 3-methoxytyrosine; homocitrulline; ornithine; succinylcarnitine	serotonin (5HT); glycerate; docosahexaenoate (DHA, 22:6n3); androsterone sulfate	
COPD vs. Control Severity	Ubhi et al.	Metabolic profiling detects biomarkers of protein degradation in COPD patients	ECLIPSE Cohort 15 Non-smokers/53 Smoker/163 COPD 64.5% males 61/57/64.2 (non-S/S/COPD)	NMR; LC-MS	Fasting Serum	COPD: 3-methylhistidine; Glutamine; Phenylalanine; 3-hydroxybutyrate; Acetoacetate GOLD IV: Trimethylamine; 3-methylhistidine; Glutamine; Phenylalanine; 3-hydroxybutyrate	COPD: N,N-dimethylglycine; LDL/VLDL; Polyunsaturated lipid; O-acetylated glycoproteins GOLD IV: N,N-dimethylglycine; 3-hydroxyisobutyrate; Isobutyrate; Isoleucine; Valine	
COPD vs. Healthy Control (Severity)	Xue et al.	Metabolomic profiling of anaerobic and aerobic energy metabolic pathways in chronic obstructive pulmonary disease	140 COPD/20 HC 78.8% males 60/52 yrs (COPD/HC)	UHPLC-Q-TOF/MS	Fasting Serum	GOLD IV: pyruvate; lactic acid	TCA Cycle	
COPD vs. Smokers	Berdyshev et al.	Ceramide and sphingosine-1 phosphate in COPD lungs	69 COPD/16 Smoker without COPD/13 Interstitial lung disease 48.4% males 66/62.8/60.5 years (S/COPD/ILD)	LC-MS	Lung tissue	ceramides (GOLD0—2) sphingosine-1 phosphate	ceramides (GOLD 3–4)	ceramide-to-S1P metabolism controlled by sphingosine kinase-1 (SphK1)
COPD vs. Smokers	Diao et al.	Disruption of histidine and energy homeostasis in chronic obstructive pulmonary disease	79 COPD/59 smokers no COPD/7 non-smokers 100% male 58.8/56.8/57.4 yrs (COPD/S/non-S)	NMR	Fasting Serum and Plasma	histamine	creatine; glycine; histidine; threonine	
COPD vs. Smokers	Gaida et al.	A dual center study to compare breath volatile organic compounds from smokers and non-smokers with and without COPD	52 Healthy non-/ex-smoker/52 COPD non-/ex-smoker/29 Healthy Smoker/ 37 COPD Smoker 52% male 35/64/43.5/61 yrs (HC/COPD non-S/HS/COPD Smoker)	Enose IMS detector GC-IMS TD-GC-APCI-MS	Exhaled Breathe	Indole; 1,6-Dimethyl-1,3,5-heptatriene; m,p-Xylene; 1-Ethyl-3-methyl benzene; Toluene; Benzene		
COPD vs. Smokers	Naz et al.	Metabolomics analysis identifies sex-associated metabotypes of oxidative stress and the autotaxin–lysoPA axis in COPD	Karolinska COSMIC cohort 38 Never-smokers/40 smokers/38 COPD 51.7% males 58.8/53/60.3 yrs (NS/S/COPD)	LC-MS	Serum	*Both Sexes*: 12-HETE; 4-HdoHE; Carnitine(C12:0); Carnitine(C14:0); Carnitine(C14:1) *Females*: 12-HETE; 4-HdoHE; Carnitine(C12:0); Dityrosine; Erythronicacid	*Both Sexes*: Leu-Pro; PC(16:1/P-18:1)	*Both Sexes*: Citrate (tricarboxylic acid) cycle; Glycerophospholipid metabolism; Pyruvate metabolism *Females*: Fatty acid biosynthesis; Sphingolipid metabolism *Males*: cAMP signaling pathway; Retrograde endocannabinoid signaling; Tryptophan metabolism
Case vs. Control (emphysema)	Ubhi et al.	Metabolic profiling detects biomarkers of protein degradation in COPD patients	ECLIPSE Cohort 41 non-emphysematous/77 Emphysematous 68.6% males 64/64 yrs (non-E/Emph)	NMR; LC-MS	Fasting Serum	3-methylhistidine; Glutamine; Phenylalanine; 3-hydroxybutyrate	Creatine; Glycine; N,N-dimethylglycine; 3-hydroxyisobutyrate; Isoleucine	
Case vs. Control (GOLD)	Ubhi et al.	Targeted metabolomics identifies perturbations in amino acid metabolism that sub-classify patients with COPD	ECLIPSE Cohort 30 smoker/30 COPD/100% male 57/65 (S/COPD)	LC-MS/MS	Fasting Serum	Beta-Aminoisobutyric acid*; 3-Methylhistidine; Aspartic acid; 1-Methylhistidine; Glutamine	alpha-Aminobutyric acid*; Proline; Aminoadipic acid; 4-Hydroxyproline; Leucine	
Case vs. Control (emphysema)	Ubhi et al.	Targeted metabolomics identifies perturbations in amino acid metabolism that sub-classify patients with COPD	ECLIPSE Cohort 21 no emphysema/38 Emphysema 100% male 65/64 yrs (non-E/Emph)	LC-MS/MS	Fasting Serum	Glutamine; Aspartic acid; 3-Methylhistidine; 1-Methylhistidine; Histidine	Tryptophan; Sarcosine; beta-Aminoisobutyric acid; Aminoadipic acid	
Case vs. Control (cachexic)	Ubhi et al.	Targeted metabolomics identifies perturbations in amino acid metabolism that sub-classify patients with COPD	ECLIPSE Cohort 30 no Cachexia/29 Cachexia 100% male 64/61 yrs (no Cach/Cach)	LC-MS/MS	Fasting Serum	Serine; Glutamine; Glutamic acid; Histidine; Asparagine	Cystathionine; Thiaproline; 1-Methylhistidine; Sarcosine; beta-Aminoisobutyric acid	
Emphysema	Bowler et al.	Plasma Sphingolipids Associated with Chronic Obstructive Pulmonary Disease Phenotypes	COPDGene Cohort *Targeted*: *n* = 129; 57% male; 63 yrs *Untargeted*: *n* = 131; 56% male; 64 yrs	LC-MS	Plasma		Ganglioside GM3 (d18:1/16:0); Sphingomyelin(d18:0/24:1(15Z)); Sphingomyelin(d18:1/14:0); Sphingomyelin(d18:1/16:0); Sphingomyelin(d18:1/16:1);	
Emphysema	Halper-Stromberg et al.	Bronchoalveolar Lavage Fluid from COPD Patients Reveals More Compounds Associated with Disease than Matched Plasma	SPIROMICS cohort 12 Non-smokers/56 Smokers/47 COPD 50.7% male 56/58/64 yrs (non-S/S/COPD)	LC-MS	BALF	leucine; lysine		Amino acid derived compounds; fatty acids; phospholipids (phosphatidylethanolamines, phosphatidylinositols, phosphatidylcholines), carnitines
Emphysema	Labaki et al.	Serum amino acid concentrations and clinical outcomes in smokers	SPIROMICS cohort: *n* = 157; 49.7% male; 53.7 yrs	NMR	Serum		tryptophan	
Emphysema	Mastej et al.	Identifying Protein–metabolite Networks Associated with COPD Phenotypes	COPDGene Cohort 426 Controls/478 COPD/92 PRISm 12 missing Spirometry 50.9% males 64.6/71.1/67.3/72 yrs (HC/COPD/PRISm/missSpiro)	LC-MS	Plasma		1-stearoyl-2-linoleoyl-GPI (18:0/18:2); androsterone glucuronide; 1-stearoyl-2-docosahexaenoyl-GPE (18:0/22:6); 1-palmitoyl-2-docosahexaenoyl-GPE (16:0/22:6); 1-palmitoyl-2-linoleoyl-GPI (16:0/18:2)	
Emphysema	Gillenwater et al.	Metabolomic Profiling Reveals Sex Specific Associations with Chronic Obstructive Pulmonary Disease and Emphysema	COPDGene Cohort: *n* = 839; 51.7% male; 67 yrs SPIROMICS Cohort: *n* = 446; 52% male; 64.5 yrs	LC-MS (Metabolon)	Plasma	*Network Module*: Amino Acids, Bile Acids, Acyl Cholines, Lysophospholipids *Network Module*: Steroids *Network Module*: Xenobiotics, Amino Acids, and TCA cycle; Amino Acids, Bile Acids, Acyl Cholines, Lysophospholipids *Network Module*: Steroids	Full Cohort: 5-hydroxylysine, isovalerate (C5), X-17357 *Males*: 5-hydroxylysine; X-17357 *Females*: 2,3-dihydroxy-2-methylbutyrate; alpha-ketoglutaramate; homocitrulline	
Emphysema	Gillenwater et al.	Plasma Metabolomic Signatures of Chronic Obstructive Pulmonary Disease and the Impact of Genetic Variants on Phenotype-Driven Modules	Discovery—COPDGene Cohort: *n* = 957; 51.2% male; 68.3 yrs Replication—COPDGene-Emory: *n* = 271; 46.9% male; 67.3 yrs Replication—SPIROMICS-Metabolon: *n* = 445; 54.8% males; 65.3 yrs Replication—SPIROMICS-UC: *n* = 76; 52.6% male; 61.6 yrs	LC-MS (Metabolon)	Plasma	tricarboxylic cycle metabolite (citrate)		
Emphysema	Diao et al.	Disruption of histidine and energy homeostasis in chronic obstructive pulmonary disease	79 COPD/59 smokers no COPD/7 non-smokers 100% male 58.8/56.8/57.4 yrs (COPD/Smoker/NS)	NMR	Fasting Serum and Plasma	creatine; histidine; 3-hydroxybutyrate; betaine; carnitine		
FEV1 (% predicted) FEV1/FVC	Bregy et al.	Real-time mass spectrometric identification of metabolites characteristic of chronic obstructive pulmonary disease in exhaled breath	22 COPD/14 controls 61.1% male 58.6/58.1 yrs (COPD/HC)	SESI-HRMS	EBC	Pyridine; 11-hydroxyundecanoic acid; (+)-γ-hydroxy-L-homoarginine; Oxo-tetradecenoic acid; Hexadecatrienoic acid; Oxo-heptadecanoic acid	2-hydroxyisobutyric acid; Aspartic acid semialdehyde; Acetohydroxybutanoic acid; 2-oxoglutaric acid semialdehyde	
FEV1 FEV1/FVC	Celejewska-Wójcik et al.	Eicosanoids and Eosinophilic Inflammation of Airways in Stable COPD	76/37 COPD/HC 68% male 65 years (mean age)	GC-MS HPLC-MS	Sputum		PGD2; 11-dehydro-TBX2	
FEV1 % predicted FEV1/FVC	Cruickshank-Quinn et al.	Metabolomics and transcriptomics pathway approach reveals outcome-specific perturbations in COPD	COPDGene Cohort: (*n* = 149); 53.0% male; 63.1 yrs	LC-MS	Plasma	PE(P-38:2); Sphinganine-1-phosphate; Eicosapentaenoyl PAF C-16; Purine	cis-7-Hexadecenoic acid methyl ester; LysoPC(16:0); Ceramide (d18:1/24:1); Octanoyl-L-carnitine; Glucosaminic acid	Endocytosis; Fc gamma R-mediated phagocytosis; Glycerophospholipid metabolism; Hippo signaling pathway; Jak-STAT signaling pathway; Lysosome; mTOR signaling pathway; Neurotrophin signaling pathway; NF-kappa B signaling pathway; Notch signaling pathway; Osteoclast differentiation; Peroxisome; Phagosome; Phosphatidylinositol signaling system; Primary immunodeficiency; Ribosome; SNARE interactions in vesicular transport; Sphingolipid metabolism; Sphingolipid signaling pathway; T cell receptor signaling pathway; Th1 and Th2 cell differentiation; Th17 cell differentiation; Autophagy; Fat digestion and absorption; Glycerolipid metabolism; Hematopoietic cell lineage
FEV1/FVC	Yu et al.	Metabolomics Identifies Novel Blood Biomarkers of Pulmonary Function and COPD in the General Population	ARIC Cohort (*n* = 2354 African Americans, 1529 European American); 39.8% male; 53.0/54.6 yrs (AA/EA) KORA cohort (*n* = 859 Europeans); 46.8% male; 53.8 yrs	LC-MS	Serum	androsterone sulfate; dehydroisoandrosterone sulfate (DHEA-S); lathosterol	3-methoxytyrosine; glycerol; oleoylcarnitine; 7-alpha-hydroxy-3-oxo-4-cholestenoate (7-Hoca); theophylline	
FEV1/FVC	Prokić et al.	A cross-omics integrative study of metabolic signatures of chronic obstructive pulmonary disease	*Discovery*: 4948 Rotterdam Study (44.2% male; 70.3 yrs); 609 Erasmus Rucphen Family study (44.2% male; 49 yrs) *Validation*: 717 Lifelines-DEEP study (43.7% male; 46 yrs); 11,498 FINRISK (47.7% male 49.7 yrs); 854 Prospective Investigation of the Vasculature in Uppsala Seniors (51.8% male; 70 yrs)	NMR	Fasting Plasma	*Discovery*: Valine	*Discovery*: GlycA	
FEV1/FVC -post	Gillenwater et al.	Plasma Metabolomic Signatures of Chronic Obstructive Pulmonary Disease and the Impact of Genetic Variants on Phenotype-Driven Modules	Discovery—COPDGene Cohort: *n* = 957; 51.2% male; 68.3 yrs Replication—COPDGene-Emory: *n* = 271; 46.9% male; 67.3 yrs Replication—SPIROMICS-Metabolon: *n* = 445; 54.8% males; 65.3 yrs Replication—SPIROMICS-UC: *n* = 76; 52.6% male; 61.6 yrs	LC-MS (Metabolon)	Plasma	1-stearoyl-2-oleoyl-GPE (18:0/18:1); propionylcarnitine (C3); ergothioneine; 3-formylindole; 1-stearoyl-2-linoleoyl-GPE (18:0/18:2)*	myristoleoylcarnitine (C14:1); decanoylcarnitine (C10); sphingomyelin (d18:2/18:1); cis-4-decenoylcarnitine (C10:1); laurylcarnitine (C12)	diacylglycerol and BCAA (leucine, isoleucine, and valine)
FEV1/FVC	Halper-Stromberg et al.	Bronchoalveolar Lavage Fluid from COPD Patients Reveals More Compounds Associated with Disease than Matched Plasma	SPIROMICS cohort 12 Non-smokers/56 Smokers/47 COPD 50.7% male 56/58/64 (NS/S/COPD)	LC-MS	BALF Plasma	BALF: Phosphatidylserine (37:3); Lophocerine; p-cresol; Phosphatidylethanolamine (38:3); Phosphatidycholine (40:6) Plasma: arginine; isoleucine; serine	BALF: Ceramide (d18:1/16:0)	Amino acid derived compounds; fatty acids; phospholipids (phosphatidylethanolamines, lysophosphatidylethanolamines, lysophosphatidylcholines, phosphatidylserines, phosphatidylinositols, phosphatidylcholines)
FEV1 % predicted	Labaki et al.	Serum amino acid concentrations and clinical outcomes in smokers	SPIROMICS cohort: *n* = 157; 49.7% male; 53.7 yrs	NMR	Serum	tryptophan		
FEV1 % predicted	Gillenwater et al.	Plasma Metabolomic Signatures of Chronic Obstructive Pulmonary Disease and the Impact of Genetic Variants on Phenotype-Driven Modules	Discovery—COPDGene Cohort: *n* = 957; 51.2% male; 68.3 yrs Replication—COPDGene-Emory: *n* = 271; 46.9% male; 67.3 yrs Replication—SPIROMICS-Metabolon: *n* = 445; 54.8% males; 65.3 yrs Replication—SPIROMICS-UC: *n* = 76; 52.6% male; 61.6 yrs	LC-MS (Metabolon)	Plasma	phosphocholine; ergothioneine; gamma-glutamyl-2-aminobutyrate; dehydroepiandrosterone sulfate (DHEA-S); 3-formylindole	N6-carboxymethyllysine; cis-4-decenoylcarnitine (C10:1); 5-dodecenoylcarnitine (C12:1); C-glycosyltryptophan; myristoleoylcarnitine (C14:1)	
FEV1 % predicted	Mastej et al.	Identifying Protein–metabolite Networks Associated with COPD Phenotypes	COPDGene Cohort 426 Controls/478 COPD/92 PRISm 12 missing Spirometry 50.9% males 64.6/71.1/67.3/72 (HC/COPD/PRISm/MissSpiro)	LC-MS	Plasma	Phosphocholine; Ergothioneine	5-hydroxyhexanoate; Palmitoleoylcarnitine (C16:1); Myristoleoylcarnitine (C14:1); Cis-4-decenoylcarnitine (C10:1); (N(1) + N(8))-acetylspermidine	
FEV1 % predicted	Diao et al.	Disruption of histidine and energy homeostasis in chronic obstructive pulmonary disease	79 COPD/59 smokers no COPD/7 non-smokers 100% male 58.8/56.8/57.4 yrs (COPD/S/non-S)	NMR	Fasting Serum and Plasma	creatine; histidine; threonine; lactate; proline; serine		
FEV1 % predicted	Balgoma et al.	Linoleic acid-derived lipid mediators increase in a female-dominated subphenotype of COPD	Karolinska COSMIC 25 COPD Smokers/10 COPD Former Smoker/40 Smokers/39 Never-Smokers 50% males 59.5/60/54/56.5 (COPD S/COPD FS/S/NS)	LC-MS/MS	BALF	*Females:* EpOMEs; DiHOMEs.	*Score*: 9,10,13-TriHOME (9,10,13-trihydroxy-11E-octadecenoic acid), 12(13)-EpOME (12[13]epoxy-9Z-octadecenoic acid), 9(10)-EpOME (9[10]-epoxy-12Z-octadecenoic acid), 9,10-DiHOME (9[10]-dihydroxy-12Z-octadecenoic acid), 12,13-DiHOME (12[13]-dihydroxy-12Zoctadecenoic acid), 12-HHTrE (12-hydroxy-5Z,8E,10E-heptadecatrienoic acid), 5-KETE (5-oxo-ETE, 5-oxo- 6E,8Z,11Z,14Z-eicosatetraenoic acid), TXB2 (thromboxane B2) and 9-KODE (9-oxo-10E,12Z-octadecadienoic acid)	
FEV1 % predicted	Balgoma et al.	Linoleic acid-derived lipid mediators increase in a female-dominated subphenotype of COPD	Karolinska COSMIC 25 COPD Smokers/10 COPD Former Smoker/40 Smokers/39 Never-Smokers 50% males 59.5/60/54/56.5 (COPD S/COPD FS/S/NS)	LC-MS/MS	BALF	PGF2α, 12-HHTrE, 12-HETE, 11(12)-EpETrE, 9,10,13-TriHOME, 5(6)-EpETrE, 11-β-PGF2α	5,6-DiHETrE, 5-HEPE, 5-HETE	
FEV1	McClay et al.	1H Nuclear Magnetic Resonance Metabolomics Analysis Identifies Novel Urinary Biomarkers for Lung Function	197 COPD/90 Smokers/105 Never-Smokers 56.4% males 64.7/57.2/56.5 (COPD/S/NS)	NMR	Plasma Urine	trigonelline; ippurate; formate		
Lung Function	Xue et al.	Metabolomic profiling of anaerobic and aerobic energy metabolic pathways in chronic obstructive pulmonary disease	140 COPD/20 HC 78.8% males 60/52 yrs (COPD/HC)	UHPLC-Q-TOF/MS	Fasting Serum	Citrate; alpha-ketogluatara; Succinate; Fumarate; Oxa	Isocitrate; Malate; Pyruvic; Lactic	
FEV1	Yu et al.	Metabolomics Identifies Novel Blood Biomarkers of Pulmonary Function and COPD in the General Population	ARIC Cohort (*n* = 2354 African Americans, 1529 European American); 39.8% male; 53.0/54.6 yrs (AA/EA) KORA cohort (*n* = 859 Europeans); 46.8% male; 53.8 yrs	LC-MS	Serum	glycine; 3-phenylpropionate (hydrocinnamate); asparagine; glutamine; serotonin (5HT)	3-(4-hydroxyphenyl)lactate; 2-methylbutyrylcarnitine (C5); alpha-hydroxyisovalerate; isoleucine; lactate	Aminoacyl-tRNA biosynthesis; Phenylalanine metabolism; Nitrogen metabolism; Alanine, aspartate and glutamate metabolism
FVC	Yu et al.	Metabolomics Identifies Novel Blood Biomarkers of Pulmonary Function and COPD in the General Population	ARIC Cohort (*n* = 2354 African Americans, 1529 European American); 39.8% male; 53.0/54.6 yrs (AA/EA) KORA cohort (*n* = 859 Europeans); 46.8% male; 53.8 yrs	LC-MS	Serum	glycine; N-acetylglycine; asparagine; glutamine; 3-phenylpropionate (hydrocinnamate)	isoleucine; 2-methylbutyrylcarnitine (C5); 3-(4-hydroxyphenyl)lactate; tyrosine; valine	Aminoacyl-tRNA biosynthesis; Phenylalanine metabolism
Exacerbation vs. Not	Celejewska-Wójcik et al.	Eicosanoids and Eosinophilic Inflammation of Airways in Stable COPD	76 COPD/37 HC 68% male 65 yrs (mean age)	GC-MS HPLC-MS	Sputum	PGD2; 12-oxo-ETE; 5-oxo-ETE		
Exacerbation vs. Not	Labaki et al.	Serum amino acid concentrations and clinical outcomes in smokers	SPIROMICS cohort: *n* = 157; 49.7% male; 53.7 yrs	NMR	Serum		O-acetylcarnitine; Lysine; 2-hydroxybutyrate; Tryptophan; Leucine	
Exacerbations	Esther et al.	Identification of Sputum Biomarkers Predictive of Pulmonary Exacerbations in Chronic Obstructive Pulmonary Disease	SPIROMICS cohort 77 healthy non-smokers/341 smokers 29reserved spirometry/562 COPD 53% male 55.4/59.6/65 yrs (HC/Smokers/COPD)	UPLC-MS	Sputum	Sialic Acid; Hypoxanthine		
Exacerbation Frequency	Cruickshank-Quinn et al.	Metabolomics and transcriptomics pathway approach reveals outcome-specific perturbations in COPD	COPDGene Cohort: (*n* = 149); 53.0% male; 63.1 yrs	LC-MS	Plasma	Carnitine (C14:2)	L-Glutamine; Pyroglutamic acid; Tryptophan; Tyrosine; Oleamide	Aminoacyl-tRNA biosynthesis; Antigen processing and presentation; Glycerophospholipid metabolism; Mineral absorption; Protein digestion and absorption; Ribosome; RNA transport
Exacerbation Frequency	Gillenwater et al.	Plasma Metabolomic Signatures of Chronic Obstructive Pulmonary Disease and the Impact of Genetic Variants on Phenotype-Driven Modules	Discovery—COPDGene Cohort: *n* = 957; 51.2% male; 68.3 yrs Replication—COPDGene-Emory: *n* = 271; 46.9% male; 67.3 yrs Replication—SPIROMICS-Metabolon: *n* = 445; 54.8% males; 65.3 yrs Replication—SPIROMICS-UC: *n* = 76; 52.6% male; 61.6 yrs	LC-MS (Metabolon)	Plasma		N,N,N-trimethyl-alanylproline betaine (TMAP)	
Exacerbation Severity	Cruickshank-Quinn et al.	Metabolomics and transcriptomics pathway approach reveals outcome-specific perturbations in COPD	COPDGene Cohort: (*n* = 149); 53.0% male; 63.1 yrs	LC-MS	Plasma	Lysine; Cholic acid; Alpha-D-glucose; Mannitol	Acetylcarnitine; Citrulline; Creatinine; L-Glutamine; L-Norvaline	ABC transporters; Aminoacyl-tRNA biosynthesis; Arginine and proline metabolism; Arginine biosynthesis; Autophagy; Glycerophospholipid metabolism; Glycine, serine and threonine metabolism; Insulin resistance; Mineral absorption; Phenylalanine, tyrosine and tryptophan biosynthesis; Protein digestion and absorption; Purine Metabolism; Retrograde endocannabinoid signaling; Sphingolipid metabolism; Sphingolipid signaling pathway
Exacerbations (Moderate & Severe)	Bowler et al.	Plasma Sphingolipids Associated with Chronic Obstructive Pulmonary Disease Phenotypes	COPDGene Cohort *Targeted*: (*n* = 129; 57% male; 63 yrs) *Untargeted*: (*n* = 131; 56% male; 64 yrs)	LC-MS	Plasma	Trihexosylceramide (d18:1/16:0); 3-O-Sulfogalactosylceramide (d18:1/16:0); Galabiosylceramide (d18:1/24:1(15Z))	Sphingosine 1-phosphate	
GOLD Stage	Gillenwater et al.	Plasma Metabolomic Signatures of Chronic Obstructive Pulmonary Disease and the Impact of Genetic Variants on Phenotype-Driven Modules	Discovery—COPDGene Cohort: *n* = 957; 51.2% male; 68.3 yrs Replication—COPDGene-Emory: *n* = 271; 46.9% male; 67.3 yrs Replication—SPIROMICS-Metabolon: *n* = 445; 54.8% males; 65.3 yrs Replication—SPIROMICS-UC: *n* = 76; 52.6% male; 61.6 yrs	LC-MS (Metabolon)	Plasma		Ergothioneine	
IL6	Diao et al.	Disruption of histidine and energy homeostasis in chronic obstructive pulmonary disease	79 COPD/59 smokers no COPD/7 non-smokers 100% male 58.8/56.8/57.4 yrs (COPD/Smoker/non-S)	NMR	Fasting Serum and Plasma	creatine; glycine; histidine; carnitine; lactate		
TNF-alpha	Diao et al.	Disruption of histidine and energy homeostasis in chronic obstructive pulmonary disease	79 COPD/59 smokers no COPD/7 non-smokers 100% male 58.8/56.8/57.4 yrs (COPD/Smoker/non-S)	NMR	Fasting Serum and Plasma	histidine; betaine; glutamine; acetylcarnitine; valine		
Subtyping	Gillenwater et al.	Multi-omics subtyping pipeline for chronic obstructive pulmonary disease	COPDGene Cohort: *n* = 1057; 50.5% male; 67.6 yrs	LC-MS (Metabolon)	Plasma	Support Vector Machine with Recursive Feature Extraction Metabolites: dehydroisoandrosterone sulfate (DHEA-S); 3-(3-amino-3-carboxypropyl)uridine; X—12,117; stearoyl sphingomyelin (d18:1/18:0); hydroxy-CMPF		Sphingomyelins
6MWD	Labaki et al.	Serum amino acid concentrations and clinical outcomes in smokers	SPIROMICS cohort: *n* = 157; 49.7% male; 53.7 yrs	NMR	Serum	tryptophan		

AA—African American; AECOPD—Acute Exacerbation of COPD; BALF—bronchoaveolar lavage fluid; Cach—cachexia; Cer—ceramide; DI-ESI-QTOF-MS—direct infusion electrospray ionization triple-quadrupole-time-of-flight mass spectrometry; EA—European American; EBC—exhaled breath condensate; Emph—emphysematous; ESI-MS—electrospray ionization—MS; FS—former smoker; GC—gas chromatography; GC-IMS—gas chromatography—ion mobility spectrometry; HC—healthy controls; HDL—high-density lipoprotein; HPLC-MS—high-performance liquid chromatography/tandem mass spectrometry; HS—healthy smoker; ILD—interstitial lung disease; IMS- ion mobility spectrometry; IMS-MCC—ion mobility spectrometer—multi-capillary column; LC—liquid chromatography; LC—lung cancer; LC-QTOF-MS—LC-quadrupole-time-of-flight-MS; LDL—low-density lipoprotein; LTD4—leukotriene D4; LTE4—leukotriene E4; missSpiro—missing spirometry measure; MS—mass spectrometry; NMR—nuclear magnetic resonance; non-E—non-emphysematous; Non-S—non-smokers; NS—never-smoker; PGD2—prostaglandin D2; PGE2—prostaglandin E2; PI—phosphoinositol; PRISm—preserved ratio impaired spirometry; S—smoker; S-COPD—Stable COPD; SESI-HRMS—secondary electrospray ionization—high-resolution MS; TB—tuberculosis-related COPD; TD-GC-APCI—MS—thermal desorption-GC-atmospheric pressure chemical ionization—MS; UHPLC-Q-TOF/MS—ultra-high-performance liquid phase series quadrupole-flight-time/secondary MS; UPLC-MS—ultra-performance liquid chromatography MS; VLDL—very-low-density lipoprotein; yrs—years3.

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
