# Peer review of "Metabolome Features of COPD: A Scoping Review"

_metabolites, 2022, doi:10.3390/metabo12070621_

Round 1

Reviewer 1 Report

N/A

Author Response

We’d like to thank the first reviewer for reviewing the manuscript.

Reviewer 2 Report

Comments on “Metabolome Features of COPD: A Systematic Review” by Godbole and Bowler.

The review is well written and the authors have considered all the possible sample types. Authors have included a total of 37 articles and various analytical platforms that have identified a variety of metabolites. I believe this review would benefit from the few suggestions listed below.

  1. Authors should exact meaning of positive and negative association, in terms of presence or absence, increased or decreased levels compared to the levels in the healthy volunteers.
  2. Table 2: Please put positive and negative associations in separate columns.
  3. Table 2: Some cells include 2 sample types, eg., serum urine. Please clarify if the same metabolites were present in both the sample types. If there is a difference, place 2 sample types in different rows.
  4. The review will definitely benefit from a model that would give a metabolomic profile of COPD in different tissues/ genders in one glance. It should be placed along with the discussion.
  5. Manuscript mentions Table 1 but Table 1 is not provided.
  6. Table 2 in the manuscript and Supplementary Table 1 appear same.
  7. Manuscript starts with Figure 2 and Figure 1 placed at the end.
  8. Some of the references are incomplete. For eg. Ref. no. 47. Please check other references also.

Reviewer 3 Report

Godbole and Bowler conducted a comprehensive review on the topic of COPD metabolomics. It provides valuable information and points out the current limitations in the research of this significant disease. Nonetheless, the method itself is not a "systematic review".

I understand that a systematic search via Google Scholar and PubMed databases was conducted. Table 1 (mentioned in line 635, which cannot be found in the manuscript) addresses some kind of risk of biases.

Yet, a typical systematic review would include at least 3 databases (using Web of Science and/or Scopus) and one "grey literature". A systematic review would also provide a very detailed methodology of inclusion, exclusion, data extraction, quality control, evidence synthesis, and so on. Importantly, it is recommended and widely accepted worldwide that some guidelines must be followed to address the potential biases of the investigators: PRISMA for reporting, QUADAS-2 to evaluate the reporting quality of included studies of primary diagnostic accuracy studies. In this journal, there has been a publication describing the reporting recommendations for metabolomics studies, that can also be used (DOI: 10.3390/metabo10020051). These are just some typical aspects of a "systematic review" type of research.

In conclusion, I cannot accept the current work to be published as a "systematic review" due to the lack of methodological robustness. However, it would be re-considered if it is a conventional narrative review assisted by a systematic literature search.

Minor points:

  • Suggestions for technical considerations for the metabolomics approach should be discussed.
  • The quality of the current statistical practice should be discussed. For example, reporting raw p-values without proper adjustments is a bad practice.
  • It may be also good to give stronger/direct suggestions for what should be done to improve the utilization of metabolomics in COPD.

Round 2

Reviewer 3 Report

The manuscript is now suitable for a formal publication.